# The Role of Body Image, Disordered Eating and Lifestyle on the Quality of Life in Lithuanian University Students

**DOI:** 10.3390/ijerph17051593

**Published:** 2020-03-02

**Authors:** Migle Baceviciene, Rasa Jankauskiene, Vaiva Balciuniene

**Affiliations:** 1Department of Physical and Social Education, Lithuanian Sports University, Sporto 6, 44221 Kaunas, Lithuania; vaibal@stud.lsu.lt; 2Institute of Sport Science and Innovations, Lithuanian Sports University, Sporto 6, 44221 Kaunas, Lithuania; rasa.jankauskiene@lsu.lt

**Keywords:** nutrition, smoking, binge drinking, physical activity, positive body image, disordered eating, quality of life, students

## Abstract

*Purpose*. The present study aimed to explore the associations between body image concerns (BI), disordered eating (DE), health-related lifestyle, and the different domains of the quality of life (QoL) in a Lithuanian sample of student-aged men and women. *Methods*. A mixed-gender sample of students (*N* = 1850, 58.8% were women, average age 21.6 ± 5.0 years) completed a series of questionnaires, including health-related lifestyles, BI, DE, and QoL. A series of simultaneous multiple linear regression analyses were conducted. Mediation analyses were performed to estimate the size of the total, direct, and indirect effects of variables in the models. *Results*. The analysis of the linear regressions demonstrated that the positive trait of body areas satisfaction was associated with the significantly enhanced QoL in all domains in both genders (for men β = 0.29–0.34; for women β = 0.26–0.33; *p* < 0.001). DE was associated with a poorer psychological QoL domain in women only (β = −0.07; *p* = 0.047). The drive for muscularity was associated with a lower QoL in men (β = −0.06–(−0.141); *p* < 0.05). Body areas satisfaction mediated the associations between body mass index and psychological and physical QoL in both genders (TLI (Tucker Lewis Index) = 0.975; CFI (comparative fit index) = 0.997; RMSEA (Root of the Mean Square Error) = 0.053). *Conclusions.* Positive traits of BI play essential roles in the QoL of student-aged women and men. The present study adds empirical evidence emphasizing the importance of integrating education about positive body image while implementing healthy lifestyle and QoL promotion programs in student-aged men and women.

## 1. Introduction

Body image concerns and disordered eating are major health problems that persist from a young age into adulthood [1]. Research suggests that sociocultural pressures lead to the internalization of the thin and/or muscular/athletic body ideal and social comparisons [2,3]. Failing to adjust to a socially stereotyped body image leads to body dissatisfaction in men and women [2]. In turn, body dissatisfaction is associated with disordered eating [1]. Body image concerns and eating pathology are associated with lower psychosocial functioning and an unhealthy lifestyle in young adults [1,4]. 

Typically, young women desire to attain a thin body image. The drive for thinness in women is associated with various health compromising weight–reduction -related strategies such as dieting, dysfunctional exercising, purging, laxative use, etc. [1]. However, the body image possessed by men is more complex compared to women, since men desire to achieve a muscular/athletic body along with low body fat [3]. These desires are associated with the drive for muscularity, which comprise muscle-gain -related attitudes and behaviors such as muscle-gain-related eating, excessive exercising, and dieting [5]. Studies in men have demonstrated that men’s athletic/muscular and thin ideal internalizations are associated with the development of pathologic eating as well [6,7]. 

Body image concerns and disordered eating are associated with lower psychological and physical quality of life (QoL) in student-aged women and men [8,9,10,11,12]. Studies exploring how body image concerns affect perceived QoL concluded that individuals with a negative body image are more likely to diet, skip meals, to develop disordered eating, to avoid socializing, and even avoid seeking medical care [13]. It was also reported that the binge eating disorder is associated with depression and later mediates the relationships between binge eating and QoL in student aged men and women [14]. Studies also demonstrated that eating disorders predict lower mental and physical QoL in students [8] and body dissatisfaction mediates the relationships between body mass index and QoL in student-aged women [9]. Increased body shape concerns and emotional eating are associated with a lower QoL in student-aged women and poorer appearance evaluation is associated with a poorer QoL in student-aged men [11]. 

In contrast, Cash and Fleming (2002) demonstrated that body image might also have a positive impact on QoL in college-aged women. It was concluded in their study that body image have a positive influence on different life domains such as feelings of personal adequacy, happiness, ability to control body weight, interactions with friends, and acceptability of a sexual partner [15]. Further studies have demonstrated that positive body image is connected to self-esteem, social support, optimism, and a greater QoL [16,17]. Thus, understanding the role of negative and positive body image on QoL of students is an important scientific issue. 

Students represent a population experiencing enormous psychological distress-related to academic requirements, financial issues, social support, and dietary intake [18,19,20,21]. Body image and body weight -related concerns have important role on the health -related behaviors of student- aged populations. Therefore, it is necessary to explore the extent to which body image concerns impact the QoL of young people in order to understand the size of the public health problem that it represents. There is a lack of studies analyzing the associations between body image and QoL in the large samples of students in different cultures [12]. These studies are important for the developing effective and culturally sensitive healthy lifestyle and quality of life promotion programs in student-aged populations. 

Further, insufficient attention has been given to QoL and lifestyle research focusing on university students [22]. Some studies have demonstrated that tobacco smoking, binge drinking, and lack of exercise are-related to a lower QoL [23,24], while insufficient sleep duration and exercising are associated with higher depression symptoms in student-aged women and men [25]. It has been concluded that the lifestyle of emerging adults is worrying [26]. Thus, fostering a healthy lifestyle and QoL in young people is an important public health issue. Given that body image concerns are one of the major distress-related problems in youth, it is important to understand the impact of the interrelationships between body image concerns, disordered eating, and lifestyle factors with the different QoL domains. A recent systematic review demonstrated that the majority of studies examining the lifestyles of student-aged women and men have been implemented in medical student samples [22]; however, it is essential to explore large samples of students who represent various study areas. Thus, the present study aimed to fill this gap. 

The overweight and obesity numbers in young people are rising worldwide [27]. It has been demonstrated that a greater body mass index is associated with body image dissatisfaction and reduced physical QoL in student-aged women and men [9]. However, there is evidence that body dissatisfaction mediates the association between the body mass index (BMI) and physical QoL [9]. However, there is a lack of studies exploring the associations between BMI, body dissatisfaction, and QoL in student-aged populations. The results of these studies might have important implications for obesity prevention and clinical practice. 

The aim of the present study was to explore the associations between body image concerns, disordered eating, and health-related lifestyle in a large sample of students of both genders of Lithuania. We expected that body image concerns and disordered eating will be associated with a poorer QoL, especially in the psychological and physical domains. Furthermore, we expected that body image concerns and disordered eating would demonstrate the greatest predictive value in the various QoL domains signaled by BMI and health–related lifestyles in women and men. Finally, we assumed that body image and disordered eating would mediate the associations between BMI and both psychological and physical QoL in men and women.

## 2. Methods 

### 2.1. Participants

Our sample (*n* = 1850) comprised undergraduate (*n* = 1641; 88.7%) and graduate (*n* = 209; 11.3%) students from various Lithuanian state universities and colleges. The sample included 763 men (41.2%) and 1087 (58.8%) women students who were enrolled in natural and agricultural (6.6%), technology (38.0%), medical and health (27.5%), or social and humanities (27.3%) study areas. Only 11 (0.6%) of the study participants did not indicate their study area. The mean age of the sample was 21.6 (standard deviation (SD = 5.0)) years.

### 2.2. Procedure

Eleven universities and four colleges participated in the cross-sectional study. The students completed self-report online questionnaires that measured their body image, disordered eating, QoL, self-esteem and health-related lifestyle. They completed the battery of questionnaires during scheduled class time (with no imposed time limit). A total of 1941 questionnaires were received, from which 56 individuals refused to participate in the survey (97.1% response rate). The final study sample of 1850 participants provided all the information necessary for statistical analysis. For the data collection, an online questionnaire was used, and the final sample contained no missing data.

### 2.3. Ethical Considerations

The researchers received ethical approval to conduct this study from the Committee for Social Sciences Research Ethics of the Lithuanian Sports University (protocol No. SMTEK-7, 13-03-2019). In accordance with the fundamental ethical and legal principles of the research, the researchers explained the purpose of the study to the students before the questionnaires were presented. The laws of anonymity, goodwill, and volunteering were followed during the survey. 

### 2.4. Measures

*Demographic data.* The participants were asked to specify their gender, age, whether they were a student of the university or college, study program, and their year of study. Demographic questions were taken from the Health Behavior among Lithuanian Adult Population questionnaire, approved for the national survey [28].

*BMI* was calculated as body mass (kg) divided by height squared (m^2^) from students’ self-reported height and weight. According to the World Health Organization, students’ BMIs were classified into four categories: underweight (<18.5 kg/m^2^), normal weight (18.5–24.9 kg/m^2^), overweight (25.0–29.9 kg/m^2^), and obese (≥30.0 kg/m^2^) [29]. The BMIs ranged from 14.0 to 47.3 (mean = 22.7, SD = 3.7) kg/m². The results showed that majority of the students were of normal body weight; however, 21.9% of men and 12.8% of women were overweight, and 3.9% and 4.2%, respectively, were obese. 

The Lithuanian version of the *Multidimensional Body–Self Relations Questionnaire–Appearance Scales (MBSRQ-AS)* [30] was used to assess the appearance-related elements of the body image construct. This instrument comprises five subscales, with responses captured on a 5-point Likert scale ranging from 1 (completely disagree) to 5 (completely agree). The appearance evaluation subscale assesses perceptions of physical attractiveness, with higher scores reflecting a higher attractiveness evaluation; the appearance orientation subscale reveals the degree of investment in one’s appearance, with higher scores indicating a greater investment; the overweight preoccupation subscale evaluates weight vigilance, dieting, fat anxiety, and eating restraint, with higher scores reflecting a greater preoccupation with being overweight; and the body area satisfaction subscale assesses satisfaction or dissatisfaction with particular areas of the body, with higher scores indicating greater body area satisfaction. The Lithuanian version of the MBSRQ-AS (LT-MBSRQ-AS) has demonstrated good validity and reliability in a student population sample [31]. In the present study, the Cronbach’s alpha values for the appearance evaluation, appearance orientation, overweight preoccupation, and body area satisfaction subscales were 0.83, 0.79, 0.73 and 0.88, respectively.

The Lithuanian version of the *Eating Disorder Examination Questionnaire 6.0 (EDE-Q 6.0)* [32] is a 28-item self-report questionnaire designed to evaluate the essential behavioral characteristics of eating disorders and eating disordered behavior. The EDE-Q 6.0 concentrates on the 28 days prior to the questionnaire and establishes two data models. First, the six open-ended questions yield frequency data on the essential behavioral characteristics of eating disorders (in terms of the number of episodes of the behavior or number of days on which the action occurred): objective binge eating, self-induced vomiting, laxative use, and excessive exercise. Second, 22 attitudinal questions across four subscales produce subscale scores that reflect the severity of the eating disorder characteristics. The responses are recorded on a 7-point Likert scale from 0 (no days) to 6 (every day), with higher scores reflecting a greater severity or higher frequency. The Lithuanian version of the EDE-Q 6.0 (LT-EDE-Q 6.0) has demonstrated good validity and reliability in a student population sample [33]. In the present study, Cronbach’s alpha for the LT-EDE-Q 6.0 was good, namely 0.94.

We used the *Drive for Muscularity Scale (DMS)* [34] to examine behaviors that reflect preoccupation with muscularity. The scale assesses individuals’ perceptions of not being muscular enough and needing to add bulk to their body frame in the form of muscle mass (irrespective of their percentage of actual muscle mass or body fat). It consists of 15 items rated a on 6-point Likert scale ranging from 1 (never) to 6 (always). Higher scores indicate more muscle development behaviors. The Lithuanian version of the DMS showed good psychometric properties [35]. In the present study, the internal consistency of the scale was good (Cronbach’s alpha = 0.89). 

The Lithuanian version of *Rosenberg’s Self-Esteem Scale* (*RSES)* [36] was used to assess self-esteem and general feelings of self-worth. The scale is composed of 10 items scored on a 4-point Likert scale ranging from 1 (strongly disagree) to 4 (strongly agree). Higher scores denote higher self-esteem. The RSES is the most widely used measure of global self-esteem. The tool demonstrated good internal consistency in the present study, with a Cronbach’s alpha of 0.89.

The Lithuanian version of *The World Health Organization Quality of Life-BREF Questionnaire (WHOQOL-BREF)* [37,38] is an abbreviated version of the World Health Organization Quality of Life-100 (WHOQOL-100) [39] self-report questionnaire. It contains 26 items and was used to assess the QoL. Two questions of the overall QoL perception and the overall understanding of health were evaluated separately. The remaining 24 items of the questionnaire comprise four domains. The 7-item (3, 4, 10, 15, 16, 17, and 18) physical health domain includes questions about dependence on medicinal substances and medical aids, pain and discomfort, activities of daily living, energy and fatigue, mobility, sleep, and rest and work capacity. The 6-item (5, 6, 7, 11, 19, and 26) psychological health domain includes questions about self-esteem, body image and appearance, negative and positive feelings, spirituality-religion, personal beliefs, and concentration. The social relations domain assesses personal relationships, social support, and sexual activity; it comprises 3 items (from 20 to 22). The 8-item (8, 9, 12, 13, 14, 23, 24 and 25) environment domain reveals information about one’s financial resources, physical safety, home environment, the possibility for recreation, opportunities for obtaining new skills and knowledge, health and social care, physical environment, and transportation satisfaction. The responses can range from 1 (very dissatisfied) to 5 (very satisfied). The scores are transformed into a scale between 0 and 100, with 0 being very poor and 100 being very good. The reliability and validity of the Lithuanian version of the WHOQOL-BREF (LT-WHOQOL-BREF) in a student population sample have been demonstrated [40]. The questionnaire was obtained from the official site of the World Health Organization [39]. The internal consistency of the domains of physical health, psychological health, social relations and the environment was 0.80, 0.85, 0.80, and 0.89, respectively. The Cronbach’s alpha for the LT-WHOQOL-BREF general scale was 0.94.

*Leisure time physical activity* was assessed using the *Leisure Time Exercise Questionnaire* (LTEQ) [41]. This instrument measures mild, moderate, and strenuous physical activity over one week. The number of bouts of mild exercise is multiplied by 3, moderate exercise by 5, and strenuous exercise by 9, all of which results in a final score of physical activity that provides a total metabolic equivalent by each intensity level. A higher score indicates higher physical activity in each of the three levels. An additional question is used to assess the frequency of vigorous exercise (rarely/never, sometimes, or often). 

*Self-rated health* was evaluated using a single question: “How would you describe your general health during the last 12 months?” There was a 4-point response scale: 1 = “poor”, 2 = “average”, 3 = “good”, and 4 = “excellent”).

*Nutrition habits* were evaluated using a food frequency questionnaire that contains 19 groups of different foods from the national survey of Health Behavior among Lithuanian Adult Population, 2014 [28]. After principal component analysis (PCA), six nutrition factors were extracted: fruits, berries, and veggies, unhealthy snacks (sweets and fast food), red meat, chicken and rice, fish, and dairy/porridge. The six factors together explained 51.48% of the total variance Kaiser-Meyer-Olkin (KMO) test = 0.72. Factor loadings from the PCA are presented in Appendix A, Table A1. In this study, only the first factor encompassing fresh vegetables, fruits, and berries consumption was used. In addition, a 5-item scale was used to evaluate unhealthy nutrition habits, such as eating while watching TV, eating in a rush, overeating, having unhealthy snacks, and eating late at night, less than 2 h before sleep. The provided response options were from “never” up to “always”. The sum of the answers for each study participant was defined as an unhealthy nutrition score and employed for further analyses as a continuous variable. The higher score indicated a higher frequency of unhealthy eating behaviors.

*Smoking and alcohol* consumption items were taken from the national survey of Health Behavior among Lithuanian Adult Population 2014 [28]. Smoking levels were assessed using a single item: “At this moment, how often do you smoke?”. The responses options were: “I have never smoked”, “I smoked some time before, but I quit”, “I occasionally smoke”, or “I smoke at least one cigarette a day.” The answers were dichotomized into yes (yes and sometimes) and no. Alcohol consumption was assessed using two items: “How often do you have a drink containing alcohol?” and “How often do you consume six standard alcohol units per one occasion?”, with the following examples of the most popular alcoholic drinks standard units. In this study, only the information about binge drinking was used. Answers were dichotomized into per cent values for less than once per month and at least once per month.

Single questions were used to indicate average *sleep duration* and *browsing the internet* times for non-educational purposes.

### 2.5. Statistical Analysis

Descriptive statistics were calculated for demographic data and each of the behaviors. The chi-square test and independent samples *t* test were performed to note any significant differences for each study variable between genders. For the comparison of nonnormally distributed data, the Mann Whitney U test was used. The internal consistency of the scales was tested using Cronbach’s alpha.

Exploratory factor analysis was performed for food frequency items using the extraction method of principal component analysis with the rotation method of varimax with Kaiser normalization. A series of simultaneous multiple linear regression analyses were conducted using age, BMI, lifestyle-related factors, body image scores, and disordered eating scores as the independent variables, and measures of psychological, physical, social, and environmental QoL domains as the dependent variables. Statistical analyses were conducted using IBM SPSS Statistics 26 (IBM Corp., Armonk, NY, USA).

Mediation analysis was conducted using AMOS version 24 (Analysis of Momentary Structure, SPSS, IBM Corp., Armonk, NY, USA) with 2000 bootstrap samples to estimate the size of the total, direct, and indirect effects and to provide 95% confidence intervals (CIs) for each effect. The significance of the direct, indirect, and total effects was assessed with chi-square tests, and the significance of the mediational paths was further confirmed through the bootstrap resampling method, with 2000 bootstrap samples and 95% bias-corrected CIs. The effects were considered significant (*p* < 0.05) if zero did not appear in the interval between the lower and the upper limits of the CIs. The goodness of fit of the models was assessed using various good fit values: goodness of fit index (GFI; 0.95 < GFI < 1.00); the adjusted goodness of fit index (AGFI; 0.90 < AGFI < 0.95); the comparative fit index (CFI; 0.95 < CFI < 1.00); the Tucker Lewis Index (TLI; 0.95 < TLI < 1.00); and the root of the mean square error of approximation (RMSEA; 0.00 < RMSEA < 0.05 indicates a good fit, while 0.05 < RMSEA < 0.08 indicates an acceptable fit).

## 3. Results

The sample characteristics are presented in Table 1. The employment rate was higher in female compared to male students. The average sleep duration was longer in men, while browsing the internet for non-educational purposes was higher in women. The physical activity score and binge drinking were more attributable to men students. Women demonstrated more frequent consumption of fresh fruits and vegetables, as well as a higher frequency of unhealthy nutrition habits and disordered eating behaviors. Body image concerns were also more prominent in women: they demonstrated higher scores of appearance orientation and overweight preoccupation, while men were more satisfied with their body areas. In addition, men rated higher on the QoL physical domain, whereas women rated higher on the social domain. There were no significant differences between the genders in the psychological and environmental QoL domains. Finally, good and excellent self-rated health was more prevalent in men compared to women.

Table 2 and Table 3 present the results from multiple linear regression analyses. These analyses largely followed the same pattern in men and women. The regression models accounted for a significant amount of variance in several of the QoL domain scores in men: psychological (R^2^ = 0.55, F = 47.74, *p* < 0.001), physical (R^2^ = 0.40, F = 26.35, *p* < 0.001), social (R^2^ = 0.32, F = 18.17, *p* < 0.001), and environmental (R^2^ = 0.32, F = 18.30, *p* < 0.001). In women, the regression models also accounted for significant variance in the psychological (R^2^ = 0.60, F = 87.46, *p* < 0.001), physical (R^2^ =0.43, F = 44.94, *p* < 0.001), social (R^2^ = 0.20, F = 14.98, *p* < 0.001), and environmental (R^2^ = 0.31, F = 27.16, *p* < 0.001) QoL domains. Not all the aspects of lifestyle-related factors were associated with the students’ QoL. However, longer sleep duration demonstrated a positive effect on the psychological and physical QoL domains in men and on the physical QoL domain in women. Moreover, more frequent consumption of fresh fruits and vegetables positively affected the women’s physical QoL domain score, whereas unhealthy nutrition habits and disordered eating behaviors were negatively-related to women’s QoL.

A higher level of body area satisfaction, self-esteem, and self-rated health played the most significant role in predicting better QoL in all domains for men and women. In addition, the drive for muscularity behaviors negatively predicted QoL in men. Better evaluated appearance and appearance-oriented behaviors were associated with higher QoL scores in both genders. In contrast, a preoccupation with being overweight decreased the assessment of the students’ QoL. Importantly, body image concerns, predominantly body area satisfaction, modified the effect of BMI on male and female students’ QoL.

In order to demonstrate whether body area satisfaction and disordered eating behaviors mediate the relationship between BMI and physical and psychological QoL domains, we conducted a series of path analyses. The model fit characteristics are presented in Table 4. Generally, men and women showed the same pattern of results. Invariance analysis did not demonstrate model differences in men and women (except for the most constrained model). Thus, the final model is presented for the entire sample.

Figure 1 presents the proposed mediation model, including the standardized path coefficients. The final model explained 4% of the body area satisfaction and 29% of disordered eating behaviors, as represented by the total EDEQ-6 score. All individual path coefficients were statistically significant and in the theoretically expected direction. A higher BMI had a negative direct effect on the body area satisfaction. Likewise, a higher level of body area satisfaction negatively predicted disordered eating, with a direct effect of −0.43 (b = −0.64; standard error [SE] = 0.029; Z = −21.70; *p* < 0.001). As expected, disordered eating behaviors had a negative effect (−0.07) on the psychological QoL domain (b = −1.14; SE = 0.30; Z = −3.80; *p* < 0.001). Conversely, better self-rated health positively predicted psychological and physical QoL domains, with direct effects of 0.17 (b = 4.11; SE = 0.44; Z = 9.27; *p* < 0.001) and 0.35 (b = 7.10; SE = 0.39; Z = 18.02; *p* < 0.001), respectively. The final model explained 42% and 34% of psychological and physical QoL domains, respectively.

The analysis of the indirect effects revealed a BMI-mediated negative effect on the psychological and physical QoL domains. Self-rated health mediated the effect between disordered eating behaviors and the psychological QoL domain. The effect between disordered eating and the physical QoL domain was fully mediated by self-rated health. A summary of the indirect effects is presented in Table 5.

## 4. Discussion

The present study aimed to explore the associations between body image, disordered eating and health-related lifestyle in a large sample of male and female students from Lithuania. We expected that body image concerns and disordered eating would be associated with a poorer QoL, especially the psychological and physical domains. Surprisingly, the first assumption was confirmed in the opposite way: the positive aspects of body image (body areas satisfaction, appearance evaluation, investment towards appearance) were associated with enhanced QoL, especially in women. In women, the majority of positive body image aspects were associated with the psychological and physical QoL domains, and disordered eating was associated with a lower psychological domain. In men, there were more associations between positive aspects of body image and physical QoL, while the drive for muscularity and overweight preoccupation decreased the QoL (mainly physical domain). These findings overlap with the findings of the other studies that demonstrated significant associations between the body image concerns, disordered eating, and lower quality of life in student-aged women and men [8,10,11,14,42]. The present study did not aim to assess positive body image; however, the findings clearly showed that positive aspects of body image (especially the satisfaction with body areas) was more strongly associated with the QoL than the aspects of negative body image (i.e., overweight preoccupation). A positive appearance evaluation does not represent positive body image [17], nevertheless, the present study demonstrated that satisfaction with body areas, a greater investment towards appearance, and a greater appearance evaluation are associated with significantly greater psychological and physical (except for appearance evaluation) domains of QoL in student-aged women. These findings support the theory of positive body image stating that a positive body image is not on the other endpoint of the same continuum with negative body image, since women might demonstrate some body image concerns, yet at the same time can report good wellbeing [17]. This is also true for men, since a greater satisfaction with body areas had much stronger associations with psychological and physical QoL than overweight preoccupation and the drive for muscularity.

This study also demonstrated that body image, particularly regarding satisfaction with parts of the body, was associated with significantly greater social QoL. Research have shown that sociocultural pressures impact body image [2]. Hence, students with greater body image satisfaction might report lower pressure from the social domain to attain a socially adored body image and therefore would evaluate social QoL more positively [2]. The novel finding of this study is that body image was positively associated with the environmental QoL domain. Perhaps students who positively evaluate their appearance demonstrate greater self-esteem and lower depression [17]; thus, they might express more positive general attitudes towards their living conditions, transportation, etc. The associations between greater overweight preoccupation and lower evaluation of environmental QoL might be mediated by a lower socioeconomic status. Studies have demonstrated that lower socioeconomic status is associated with greater body image concerns [43]. However, socioeconomic status and depression were not assessed in the present study, and this factor can be considered as a limitation. Future studies should include those variables in similar research. 

We also observed no associations between disordered eating and the psychological, physical, and social QoL domains in men. The lower associations between the disordered eating and QoL in men compared to women might be explained by the fact that the absolute number of men in the present sample was healthy and the average mean score of disordered eating in men was low [33]. However, we observed the tendency that the drive for muscularity was associated with a lower QoL in all domains. This novel finding provides evidence that decreasing the drive for muscularity in men might be an important part of health promotion programs that tackle a healthy lifestyle and good QoL. 

Next, we expected that body image and disordered eating would show the greatest predictive value in the various QoL domains in women and men after controlling by BMI and health -related lifestyle. The linear regression analyses fully confirmed this hypothesis. While some aspects of the health-related behavior (sleep duration, physical activity and the consumption of fresh fruits and vegetables) were associated with greater physical and psychological QoL, in general, the greatest predictive value for the students’ QoL emerged from positive aspects of body image, disordered and unhealthy eating (in women), and the drive for muscularity (in men). These findings have important implications. 

This study adds empirical data that demonstrates the importance of integrating biomedical and psychosocial approaches in healthy lifestyle promotion and obesity prevention programs for students [44,45]. Further, this study overlaps the rich findings of studies demonstrating the associations between the positive body image and the greater quality of life in young people [17]. The present study clearly demonstrate that it is not only important to decrease body image concerns, but it is necessary to develop positive body image in preventive programs tackling the healthy lifestyles and QoL in student–aged women and men. 

Finally, we assumed that body image and disordered eating would mediate the associations between the BMI and psychological and physical quality of life. This hypothesis was fully confirmed. Our path model showed an excellent fit to the data and demonstrated that body area satisfaction mediated the associations between BMI and both the physical and psychological QoL domains. Furthermore, body area satisfaction and disordered eating mediated the associations between BMI and the psychological QoL domain. This was true for both women and men. This finding adds to the knowledge that the negative impact of overweight or obesity on the psychological and physical QoL domains might be reduced by developing healthy body image in young women and men. Thus, preventive and clinical programs tackling overweight and obesity in student-aged populations should integrate the development of a positive body image [17]. 

The present study demonstrated that students’ lifestyles are not sufficiently healthy and presented some important gender differences. Men indicated spending fewer hours browsing the internet, higher physical activity, and a more positive body image. In addition, despite men having reported more frequent binge drinking and lower consumption of fruits and vegetables, their unhealthy nutrition score was lower compared to that of women. These findings overlap with the results from other studies [26,31,46,47,48]. However, men more frequently reported excellent/good self-rated health compared to women. Other studies demonstrated that the lower self-reported health in women depends on some important confounders, including lower physical activity and more frequent health complaints and illness periods [47]. However, the present study demonstrated that self-rated health is an important indicator of the QoL in men and women. 

The present study is cross-sectional; therefore, conclusions about the directions of the variables are limited. We strongly recommend longitudinal studies that explore the associations between health-related lifestyle, body image, and QoL. Future studies might also benefit by using the measures of body appreciation (the main trait of positive body image) [17,49]. The use of body image quality of life inventory is also recommended [16] in future studies exploring the impact of body image on the different domains of quality of life in student- aged populations. 

One of the strengths of the study is the solid sample of both gender students representing various study areas. To our knowledge, it is one of the first studies that has explored the associations between body image, disordered eating, and different QoL domains in Eastern Europe. These data should be important for health education and promotion programs. Obesity numbers in Lithuania are rising [50]; unfortunately, the implementation of obesity and body image concern prevention programs is very limited in universities and lacks methodological support. This study should fill this gap. Finally, the present study adds empirical evidence emphasizing the importance of integrating education about a positive body image while implementing healthy lifestyle and QoL promotion programs in student-aged men and women.

## 5. Conclusions

The present study demonstrated that greater body image was positively associated with the psychological and physical QoL domains in student-aged women and men. Further, disordered eating was associated with a poorer psychological domain in women. Positive traits for body image (body areas satisfaction) played essential roles in the QoL of both genders. Body area satisfaction mediated the associations between BMI and the psychological and physical QoL domains of student-aged men and women. The present study adds empirical evidence emphasizing the importance of integrating education about a positive body image while implementing a healthy lifestyle and QoL promotion programs in student-aged men and women.

## Figures and Tables

**Figure 1 ijerph-17-01593-f001:**
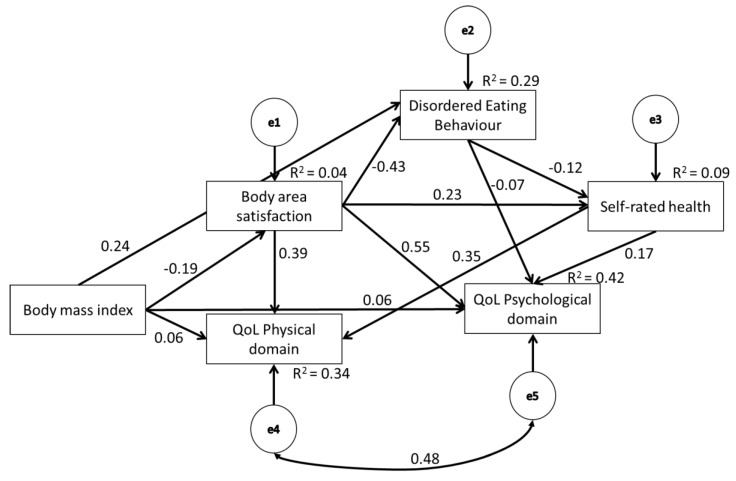
The final path model (*n* = 1850). Note: all standardized path coefficients are significant (*p* < 0.05); QoL—quality of life.

**Table 1 ijerph-17-01593-t001:** Comparison of the study variables for male and female students (*n* = 1850).

Variables	Men (*n* = 763)		Women (*n* = 1087)		*p*	d
	M	SD	M	SD		
*Sociodemographic*						
Age, years	20.4	3.1	22.4	5.8	<0.001	−0.4
Employment, %	26.3		40.9		<0.001	-
*Lifestyles*						
Sleep time duration, hours	7.3	1.1	7.2	1.1	0.059	0.09
Browsing internet duration, hours	3.0	1.9	3.1	1.8	0.045	−0.05
Physical activity score	74.1	44.0	56.6	40.2	<0.001	0.4
Binge drinking, %	29.1		16.9		<0.001	-
Smoking, %	35.5		33.9		0.459	-
*Nutrition*						
Having breakfast, %	71.2		73.4		0.287	-
Daily fresh vegetable consumption frequency, %	16.0		27.0		<0.001	-
Daily fresh fruit/berry consumption frequency, %	7.2		17.2		<0.001	-
Unhealthy nutrition score	13.7	2.9	14.3	3.0	<0.001	−0.2
EDEQ-6	0.8	0.8	1.5	1.2	<0.001	−0.6
*Body image*						
MBSRQ-AS: Appearance Evaluation	3.2	0.7	3.2	0.8	0.787	0.1
MBSRQ-AS: Appearance Orientation	3.1	0.6	3.6	0.6	<0.001	−0.8
MBSRQ-AS: Body Area Satisfaction	3.4	0.8	3.3	0.8	<0.001	0.1
MBSRQ-AS: Overweight Preoccupation	1.9	0.8	2.4	0.9	<0.001	−0.6
Drive for Muscularity	2.7	1.1	-	-	-	-
*Quality of life*						
WHOQOL-Psychological domain	59.4	19.7	59.5	17.9	0.954	−0.005
WHOQOL-Physical domain	70.5	16.3	68.7	15.1	0.02	0.1
WHOQOL-Social domain	59.0	24.5	64.8	22.1	<0.001	−0.2
WHOQOL-Environmental domain	65.3	19.6	66.7	15.5	0.109	−0.07
*Health and self-esteem:*						
Body mass index, kg/m^2^	23.4	3.6	22.1	3.6	<0.001	0.4
Self-esteem	29.4	5.9	29.6	6.1	0.385	−0.03
Excellent/good self-rated health, %	75.4		60.3		<0.001	-

M = mean; SD = standard deviation; *p* = level of significance; d = Cohen’s coefficient; MBSRQ-AS = Multidimensional Body-Self Relations Questionnaire-Appearance Scales; WHOQOL = World Health Organization Quality of Life Questionnaire; EDEQ-6.0 = Eating Disorders Examination Questionnaire.

**Table 2 ijerph-17-01593-t002:** Variables associated with different domains of the quality of life in female students (*n* = 1087).

Variables	Psychological Domain			Physical Domain			Social Domain			Environmental Domain		
	B	Β	*p*	B	β	*p*	B	Β	*p*	B	β	*p*
*Sociodemographic*												
Age	−0.105	−0.034	0.146	−0.131	−0.051	0.07	−0.359	−0.095	0.004	−0.062	−0.023	0.451
Employment	1.153	0.032	0.146	0.188	0.006	0.815	−1.023	−0.023	0.463	−1.386	−0.044	0.127
*Lifestyles*												
Sleep time duration	0.585	0.036	0.077	1.752	0.126	<0.001	0.176	0.009	0.759	0.383	0.027	0.307
Browsing internet duration	−0.040	−0.001	0.851	0.170	0.020	0.427	−0.020	−0.002	0.958	0.316	0.036	0.191
Physical activity score	0.013	0.028	0.348	−0.008	−0.020	0.410	−0.003	−0.005	0.872	−0.002	−0.006	0.812
Smoking	−0.311	−0.020	0.348	−0.244	−0.019	0.463	−0.363	−0.019	0.528	0.061	0.005	0.870
Binge drinking	0.004	0.000	0.994	−0.921	−0.048	0.054	0.181	0.007	0.827	−0.671	−0.034	0.214
*Nutrition*												
Having breakfast frequency	−0.316	−0.018	0.386	0.408	0.027	0.264	−0.162	−0.007	0.798	0.088	0.006	0.831
Fresh fruits and vegetables consumption	0.608	0.034	0.110	1.218	0.081	0.001	0.203	0.009	0.759	0.575	0.037	0.183
Unhealthy nutrition score	−0.363	−0.060	0.004	−0.373	−0.074	0.003	−0.344	−0.046	0.118	−0.391	−0.075	0.006
Disordered eating behaviors	−0.976	−0.068	0.047	0.560	0.046	0.255	−0.371	−0.021	0.664	0.698	0.056	0.210
*Body image*												
Appearance evaluation	2.839	0.133	<0.001	−0.030	−0.002	0.964	−0.243	−0.009	0.835	0.364	0.020	0.633
Appearance orientation	1.736	0.055	0.014	2.231	0.084	0.002	1.678	0.043	0.172	4.439	0.161	<0.001
Body area satisfaction	7.489	0.320	<0.001	5.140	0.260	<0.001	7.140	0.247	<0.001	6.629	0.326	<0.001
Overweight preoccupation	0.288	0.015	0.606	−1.403	−0.088	0.012	0.886	0.038	0.362	−1.921	−0.117	0.002
*Health and self-esteem*												
Body mass index	0.308	0.063	0.006	0.336	0.081	0.003	0.621	0.102	0.002	0.393	0.092	0.002
Self-rated health	3.439	0.141	<0.001	7.628	0.371	<0.001	2.525	0.084	0.005	3.541	0.167	<0.001
Self-esteem	1.013	0.345	<0.001	0.366	0.147	<0.001	0.897	0.247	<0.001	0.460	0.430	<0.001
*Model summary*	R = 0.772; R^2^ = 0.596			R = 0.657; R^2^ = 0.431			R = 0.449; R^2^ = 0.202			R = 0.560; R^2^ = 0.314		

Note: B = unstandardized regression coefficient, β = standardized regression coefficient, *p* = level of significance.

**Table 3 ijerph-17-01593-t003:** Variables associated with different domains of the quality of life in male students (*n* = 763).

Variables	Psychological Domain			Physical Domain			Social Domain			Environmental Domain		
	B	β	*p*	B	β	*p*	B	β	*p*	B	β	*p*
*Sociodemographic*												
Age	0.238	0.037	0.183	0.137	0.026	0.417	−0.003	0.000	0.992	−0.037	−0.006	0.864
Employment	−0.072	−0.002	0.954	−1.382	−0.037	0.250	2.946	0.053	0.128	−2.374	−0.053	0.125
*Lifestyles*												
Sleep time duration	1.667	0.095	<0.001	1.056	0.073	0.015	−0.113	−0.005	0.870	−0.292	−0.017	0.599
Browsing internet duration	0.020	0.002	0.941	−0.260	−0.030	0.303	−0.167	−0.013	0.681	0.454	0.044	0.162
Physical activity score	0.014	0.031	0.228	−0.023	−0.061	0.044	0.007	0.013	0.677	−0.028	−0.062	0.055
Smoking	−0.019	−0.001	0.968	−0.059	−0.004	0.894	0.720	0.034	0.312	0.317	0.019	0.578
Binge drinking	0.131	0.006	0.824	0.181	0.010	0.745	0.443	0.017	0.622	−0.611	−0.029	0.394
*Nutrition*												
Having breakfast frequency	1.036	0.052	0.052	0.181	0.011	0.720	−0.131	−0.005	0.872	1.088	0.055	0.095
Fresh fruits and vegetables consumption frequency	−0.584	−0.027	0.299	0.388	0.022	0.468	0.634	0.024	0.461	0.639	0.030	0.354
Unhealthy nutrition score	0.121	0.018	0.505	−0.171	−0.030	0.322	0.502	0.059	0.072	−0.159	−0.023	0.476
Disordered eating behaviors	−0.659	−0.028	0.394	1.159	0.060	0.115	−1.027	−0.035	0.385	2.104	0.090	0.026
*Body image:*												
Appearance evaluation	3.555	0.131	<0.001	1.108	0.050	0.205	2.436	0.072	0.084	−0.487	−0.018	0.665
Appearance orientation	0.410	0.012	0.668	1.863	0.067	0.041	−0.400	−0.009	0.785	2.673	0.079	0.023
Body area satisfaction	7.477	0.307	<0.001	6.731	0.334	<0.001	8.835	0.291	<0.001	8.296	0.342	<0.001
Overweight preoccupation	−0.441	−0.017	0.586	−2.667	−0.125	0.001	1.226	0.038	0.323	−3.259	−0.127	0.001
Drive for muscularity	−0.894	−0.052	0.056	−0.917	−0.064	0.039	−1.579	−0.073	0.028	−2.419	−0.141	<0.001
*Health and self-esteem*												
Body mass index	0.355	0.065	0.024	0.333	0.074	0.026	0.195	0.029	0.418	0.332	0.061	0.084
Self−rated health	2.532	0.103	<0.001	4.919	0.242	<0.001	3.443	0.112	0.001	1.844	0.075	0.025
Self-esteem	1.185	0.353	<0.001	0.469	0.169	<0.001	0.846	0.203	<0.001	0.779	0.233	<0.001
*Model summary*	R = 0.741; R^2^ = 0.550			R = 0.634; R^2^ = 0.403			R = 0.563; R^2^ = 0.317			R = 0.565; R^2^ = 0.319		

Note: B = unstandardized regression coefficient, β = standardized regression coefficient, *p* = level of significance.

**Table 4 ijerph-17-01593-t004:** Fit of the path models and invariance analysis between genders in a student sample (*n* = 1850).

*Models*	χ^2^	df	GFI	AGFI	TLI	CFI	RMSEA
Women (*n* = 1087)	0.970	2	1.000	0.997	1.004	1.000	0.000
Men (*n* = 763)	1.699	2	0.999	0.992	1.002	1.000	0.000
All (*n* = 1850)	12.365	2	0.998	0.977	0.975	0.997	0.053
*Invariance analysis*							
Unconstrained model	2.670	4	1.000	0.995	1.003	1.000	0.000
Structural weights	143.748	15	0.976	0.932	0.923	0.961	0.068
Structural covariances	143.799	16	0.976	0.936	0.928	0.962	0.066
Structural residuals	227.756	22	0.964	0.931	0.916	0.938	0.071

χ^2^ = chi-square; df = degrees of freedom; GFI = goodness of fit index; AGFI = adjusted goodness of fit index; CFI = comparative fit index; TLI = Tucker Lewis Index; RMSEA = root of the mean square error of approximation.

**Table 5 ijerph-17-01593-t005:** Summary of the mediation analyses that tested the indirect effect of body image and disordered eating behaviors on psychological and physical quality of life aspects (*n* = 1850).

Paths	B (95% CI)	*p*	Mediation
BMI → disordered eating behaviours	0.026 (0.019 to 0.033)	0.001	Partial
BMI → self−rated health	−0.017 (−0.022 to 0.013)	0.001	Full
BMI → psychological QoL domain	−0.434 (−0.541 to −0.334)	0.001	Partial
BMI → physical QoL domain	−0.719 (−0.880 to −0.562)	0.001	Partial
Body area satisfaction → self−rated health	0.051 (0.028 to 0.075)	0.001	Partial
Body area satisfaction → psychological QoL domain	1.863 (1.350 to 2.400)	0.001	Partial
Body area satisfaction → physical QoL domain	1.969 (1.578 to 2.368)	0.001	Partial
Disordered eating behaviors → physical QoL domain	−0.576 (−0.836 to −0.319)	0.001	Full
Disordered eating behaviors → psychological QoL domain	−0.334 (−0.513 to −0.173)	0.001	Partial

B = unstandardised effect coefficient; 95% CI = bootstrapped 95% confidence intervals for the unstandardised effect; *p* = two-tailed significance; BMI = body mass index; QoL = quality of life.

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
