# Peer review of "The Role of Body Image, Disordered Eating and Lifestyle on the Quality of Life in Lithuanian University Students"

_ijerph, 2020, doi:10.3390/ijerph17051593_

Round 1
Reviewer 1 Report
- It is unclear from the title and abstract where this study was conducted.
- I would suggest revising the title to say e.g. "The role of body image, disordered eating and lifestyle on the quality of life of Lithuanian University students".
- The study design must be clearly stated in the abstract.
- Please highlight the most significant P values in the abstract.
- Line 33-34: this reference is old, please update.
- I strongly suggest that the authors reconsider the use of the words “men and women” throughout and instead use the words “male and female”, "male/female students" or "student-aged men/women". The terminology for this term should be consistent throughout the manuscript.
- Line 38-39: What about women? They express a desire to be thin. Please explain.
- Line 41: I would suggest using "study" instead of "studies". Please also check throughout.
- Line 47-60: The authors reviewed some studies on the topic but the age ranges were unclear.
- The word "studies" throughout the paper is vague. Please define the study design used in these studies.
- Line 48: Please define "unhealthy lifestyles" here.
- My primary concern in introduction is that a conceptual rationale for the study aim is lacking. What is the conceptual basis for suggesting the potential influence of body image concerns/disordered eating on QoL, particularly among University students? Why it is important to examine body image concerns/disordered eating as mediators of the relationship between BMI and health-related QoL in men and women?
- It is unclear what study design was used and how the missing data were treated.
- The inclusion/exclusion criteria should be clearly defined.
- Line 91: What instruments used to collect demographic data? Are they valid?
- Line 93: Please clarify how weight and height were conducted?
- Line 171; 181: Please add reference(s) here.
- It would be helpful to have a table describes the results of exploratory factor analysis.
- The first paragraph in discussion should be enlarged. The results should be discussed deeply. Please also use of subheadings in this section because of the large number of results that are presented. I also suggest that there should be inclusion of more recent literature throughout the discussion.
- All the references should be formatted using the journal guideline.
Author Response
Dear Reviewer,
Thank you very much for your work and your time helping to improve the quality of our manuscript. Please find the comment-answer table attached.

Reviewer 2 Report
This is an interesting and well-written manuscript reporting a thoroughly performed and reasonably sized study with 1,850 subjects. The introduction covers the literature very well and expounds the scientific problem comprehensively. The methods section describes the scientific approach appropriately. The results section is suitable as well as enlightening, and the results are well discussed.
I have only two suggestions for improvement:
- The authors write in the abstract: "The present study aimed to explore the role of body image (BI), disordered eating (DE) and health-related lifestyle on the different domains of the quality of life (QoL) in student-aged men and women. " However, it is unclear, if someone reads only the abstract, whether "body image" means a positive body-image or body-image problems. This needs to be clarified. If the abstract then becomes too long, other information must be deleted. But the abstract should be self-explanatory.
- In the Demographic data and Discussion sections, it would also be interesting an analysis on underweight subjects. Elaborating further on the theme of drive for muscularity and orthorexia, fear of gaining weight, low BMI and anorexia on both male and female subjects and the effects of such alterations on QoL.
Author Response
Dear Reviewer,
Thank you very much for your comments. Please find the comment-answer table attached.

Round 2
Reviewer 1 Report
Dear authors,
The revision improved your paper a lot. Congratulation.